# Incidence of cough from acute exposure to fine particulate matter (PM$_{2.5}$) in Madagascar: A pilot study

**Alexandra J. Zimmer**[1,2☯], **Lai Yu Tsang**[3☯], **Gisèle Jolicoeur**[4], **Bouchra Tannir**[4], **Emmanuelle Batisse**[1], **Christine Pando**[3], **Gouri Sadananda**[5], **Jesse McKinney**[3,6], **Ideal Vincent Ambinintsoa**[6], **Roger Mario Rabetombosoa**[6], **Astrid M. Knoblauch**[7,8,9], **Niaina Rakotosamimanana**[7], **Ryan Chartier**[10], **Alina Diachenko**[4], **Peter Small**[3], **Simon Grandjean Lapierre**[2,4,7,11] *

**1** Department of Epidemiology, Biostatistics and Occupational Health, McGill University, Montreal, Canada, **2** McGill International TB Centre, McGill University, Montreal, Canada, **3** Global Health Institute, Stony Brook University, Stony Brook, New York, United States of America, **4** Immunopathology Axis, Centre de Recherche du Centre Hospitalier de l'Université de Montreal, Montreal, Canada, **5** Department of Medicine, Case Western Reserve University, Cleveland, Ohio, United States of America, **6** Centre ValBio Research Station, Ranomafana, Madagascar, **7** Mycobacteriology Unit, Institut Pasteur de Madagascar, Antananarivo, Madagascar, **8** Department of epidemiology and public health, Swiss Tropical and Public Health Institute, Allschwil, Switzerland, **9** Department of Public Health, University of Basel, Basel, Switzerland, **10** RTI International, Research Triangle Park, North Carolina, United States of America, **11** Department of Microbiology, Infectious Diseases and Immunology, Université de Montréal, Montreal, Canada

☯ These authors contributed equally to this work.
* Simon.grandjean.lapierre@umontreal.ca

**Data Availability Statement:** The cough screening was done using R software (version 4.3.0) and our code is publicly available on GitHub (https://github.com/alexjzimmer/cough-screening). The cough,

## Abstract

Prolonged exposure to fine particulate matter (PM$_{2.5}$) is a known risk to respiratory health, causing chronic lung impairment. Yet, the immediate, acute effects of PM$_{2.5}$ exposure on respiratory symptoms, such as cough, are less understood. This pilot study aims to investigate this relationship using objective PM$_{2.5}$ and cough monitors. Fifteen participants from rural Madagascar were followed for three days, equipped with an RTI Enhanced Children's MicroPEM PM$_{2.5}$ sensor and a smartphone with the ResApp Cough Counting Software application. Univariable Generalized Estimating Equation (GEE) models were applied to measure the association between hourly PM$_{2.5}$ exposure and cough counts. Peaks in both PM$_{2.5}$ concentration and cough frequency were observed during the day. A 10-fold increase in hourly PM$_{2.5}$ concentration corresponded to a 39% increase in same-hour cough frequency (incidence rate ratio (IRR) = 1.40; 95% CI: 1.12, 1.74). The strength of this association decreased with a one-hour lag between PM$_{2.5}$ exposure and cough frequency (IRR = 1.21; 95% CI: 1.01, 1.44) and was not significant with a two-hour lag (IRR = 0.93; 95% CI: 0.71, 1.23). This study demonstrates the feasibility of objective PM$_{2.5}$ and cough monitoring in remote settings. An association between hourly PM$_{2.5}$ exposure and cough frequency was detected, suggesting that PM$_{2.5}$ exposure may have immediate effects on respiratory health. Further investigation is necessary in larger studies to substantiate these findings and understand the broader implications.

PM2.5 and participant characteristics are publicly available on Synapse.org (SynID: syn59596075).

**Funding:** The authors received no specific funding for this work.

**Competing interests:** The authors have declared that no competing interests exist.

## Introduction

Air pollution is a complex mixture of particles and gasses, some of which are known to be harmful to human health. In 2019, the Global Burden of Disease Study reported that air pollution was responsible for over 6 million deaths globally [1]. Fine particulate matter (particles with a diameter of less than 2.5 μm; $PM_{2.5}$) is a major component of air pollution and $PM_{2.5}$ are of particular concern as they can be inhaled deep into the lungs causing local inflammation and oxidative stress [2].

Previous research has studied the association between $PM_{2.5}$ exposure and respiratory outcomes, including cough. Traditionally, these studies have evaluated such associations using average daily pollutant exposure and cough frequency per day as the unit of analysis [3–7]. However, other evidence suggests that the effect of air pollution on health outcomes can manifest more acutely, occurring within hours of exposure to pollutants [8–10]. For instance, a study in Japan found that peak expiratory flow (PEF) was negatively associated with increased hourly concentrations of $PM_{2.5}$ with a lag of 0, 1, 2, and 3 hours prior to measuring PEF [9]. In China, another study found that increased $PM_{2.5}$ concentrations were associated with an immediate (same hour) increase in all-cause morbidity (based on the number of emergency ambulance calls per hour) at the population level [8].

However, a major limitation of these studies is the reliance on subjective assessment tools to measure cough frequency, such as cough diaries or questionnaires [11–13]. These methods are prone to error and recall bias. Recent advances in objective cough assessment tools, including smartphone-based cough recording applications, have revolutionized the analysis of cough patterns and frequency [14–16]. These innovative tools provide a more accurate evaluation of the relationship between respiratory symptoms and cough-inducing agents, including environmental irritants. Nonetheless, the potential of these novel assessment tools to analyze the impact of environmental pollution, particularly $PM_{2.5}$, on cough frequency remains unexplored.

Objective cough monitors are increasingly being validated for the clinical management of respiratory diseases [14, 17]. Therefore, by understanding how $PM_{2.5}$ influences cough, we can adapt these devices to ensure their reliable performance even in the face of challenging environmental conditions, such as high air pollution.

In this pilot longitudinal cohort study, we aimed to provide some preliminary findings by examining the association between hourly personal $PM_{2.5}$ exposure levels and hourly cough rates among individuals living in Madagascar. We assessed real-time personal exposure to $PM_{2.5}$ using a small, lightweight $PM_{2.5}$ sensor. Additionally, we used a smartphone-based cough recording application to objectively measure cough frequency. By leveraging objective monitoring tools, our study aims to describe the acute association between air pollutants and respiratory outcomes.

## Materials and methods

### Study design and participants

A longitudinal cohort study was conducted between June 3rd and July 24th, 2018, in the Androrangavola commune of the Ifanadiana district in south-eastern Madagascar. Like other Malagasy remote and rural communities, this commune has a high prevalence of exposure to air pollution, primarily from tobacco smoke and indoor use of solid fuels for cooking [18]. Individuals 15 years and older with self-reported cough were eligible to participate in this study. Participants were enrolled through outreach case finding visits of the National Tuberculosis Program which used cough as a symptom to trigger tuberculosis (TB) testing, however

TB diagnosis was not an inclusion or exclusion criteria. Participants were instructed to maintain normal activities while wearing a neck pouch containing a $PM_{2.5}$ pollution monitor and iPhone cough monitor. Monitors were positioned in the breathing zone approximately 25–50 cm away from their mouth (S1 Fig). Participants were asked to always wear the pouch except when bathing, sitting, or sleeping, when they were instructed to keep the device nearby (within 1 meter, when possible). Participants were followed longitudinally for up to 3 days.

## Personal $PM_{2.5}$ exposure assessment

We assessed real-time personal exposure to $PM_{2.5}$ using a lightweight (150g) wearable Enhanced Children's MicroPEM (ECM) personal exposure monitor (RTI International, Research Triangle Park, NC, USA) [19]. Prior validation studies have demonstrated good agreement between this sensor and other $PM_{2.5}$ monitors [20–22]. Briefly, the ECM sensor measured real-time $PM_{2.5}$ concentrations using a light-scattering nephelometer as well as collected the sampled $PM_{2.5}$ on an internal Teflon filter (Measurement Technologies Laboratories., Minneapolis, MN, USA). All filters were pre- and post-weighed using an ultramicrobalance (Mettler Toledo UMX2) in an environmentally controlled (21˚C, 35% relative humidity [RH]) gravimetric facility. The filter masses were corrected using the mean field blank filter mass (N = 13, 0.6μg) to account for any potential contamination acquired by transit to and from the field site. All individual real-time $PM_{2.5}$ data were post-corrected using the corresponding gravimetrically determined filter concentration to improve accuracy, and an RH-correction was applied. The ECM logged $PM_{2.5}$ concentrations every 10 seconds for the first 30 seconds of every minute (30/30 cycling) at a nominal flow rate of 0.30 L/min. The ECM also collected real-time temperature measures at the same rate. The ECM's Li-ion battery was fully charged prior to giving the device to the participant to ensure continuous recording for 72h. The limit of detection for the real-time $PM_{2.5}$ was approximately 1 μg/m$^3$. Values below 1 μg/m$^3$ were replaced as half the limit of detection (0.5 μg/m$^3$).

## Digital cough recording

Cough sounds were recorded using the ResApp Cough Counting Software (ResApp Health, Brisbane, Australia) application on iPhone 6 smartphones (Apple, Cupertino, USA) [23]. ResApp Cough Counting Software uses an artificial intelligence algorithm to continuously monitor and record short "explosive sounds" corresponding to cough events. ResApp Cough Counting Software has been used as a cough data collection tool in prior studies [24]. All sound recordings were stored on secured servers hosted by ResApp Health and transferred to the research team. Sounds were listened to independently by two observers (BT and GJ) to identify cough sounds and exclude non-cough or ambiguous sounds. This cough screening was done using R software (version 4.3.0). The code is publicly available on GitHub (https://github.com/alexjzimmer/cough-screening). The cough, $PM_{2.5}$ and participant characteristics are publicly available on Synapse.org (SynID: syn59596075).

## Statistical analysis

For each participant, we calculated hourly $PM_{2.5}$ exposure by first, transforming PM2.5 concentrations using a log base 10 ($\log_{10}$) transformation and then averaging the intervals of $PM_{2.5}$ measurements. Coughs per hour was the primary outcome. For both, $PM_{2.5}$ exposure and cough rate, hours were defined based on 24-hour clock times (e.g., 9:00:00am to 8:59:59am). Only overlapping time interval for cough and pollution data were used the analysis. The first and last hours of $PM_{2.5}$ and cough recordings were excluded when there was less than 30 minutes of recording. For the first and last hours of recording, if cough monitoring

lasted between 30–59 minutes of a full hour, a prorated hourly cough rate was calculated. These prorated hourly cough rates were rounded to the nearest integer. No intermediary hourly cough recordings were pro-rated. $PM_{2.5}$ concentrations were not pro-rated.

Descriptive summary statistics of socio-demographic variables (age, sex, smoking, tuberculosis status, and household size), hourly $PM_{2.5}$ concentrations, hourly cough rates, and observation time interval were computed for each participant. Individual hourly $PM_{2.5}$ and cough trajectories were presented graphically. We investigated the association between log-10 transformed hourly $PM_{2.5}$ and hourly cough frequency using generalized estimating equation (GEE) with negative binomial distribution for outcome [25]. The negative binomial distribution was selected over the Poisson distribution to account for overdispersion of the outcome. GEE models were tested under various correlation structures, including independent, exchangeable, and first-order and second-order auto-regressive (both AR1 and AR2). The AR1 structure was selected as the best-fitting structure using the quasi-likelihood information (QIC) criterion [26]. We investigated the association between hourly $PM_{2.5}$ concentrations under three different lag scenarios in separate models: lag0, lag1, and lag2 respectively represent the association between the hourly cough rate during the same hour, the subsequent hour, and two hours after the $PM_{2.5}$ exposure measurement. The model adjustment for socio-demographic characteristics (sex, age, occupation etc.) was not performed due to small number of participants (n = 15).

To account for the previously reported natural diurnal variation in cough frequencies [27], we performed a sensitivity analysis that included an independent categorical variable dividing 24h into quarters: 0:00–5:59, 6:00–11:59, 12:00–17:59, and 18:-23:59. A second sensitivity analysis was performed to account for the possibility of participants not having their phone nearby to record coughs (e.g., if the participant forgot the phone at home). For this, we excluded hours of $PM_{2.5}$ and cough data with 4h+, 6h+, and 8h+ of subsequent recording where zero coughs per hour were recorded (S2 Fig). A final sensitivity analysis was performed, excluding raw $PM_{2.5}$ measurements that were greater than 15,000 μg/m3 (the upper limit of detection).

Both smoking status and TB status are known to influence cough [14, 28]. To explore potential effect modification according to strata of these predictors, we performed stratified analyses within levels of TB status (TB positive and TB negative) and smoking status (ever smoked and never smoked). The stratified analyses were only applied to the lag0 model (S1 Table).

Estimates extracted from GEE represent incident hourly cough rate ratios (IRR) for each 10-fold increase in hourly $PM_{2.5}$ concentrations. All statistical analyses were performed using Stata version 18 (StataCorp LP, College Station, Texas). Significance was set at 0.05, and all tests were 2-tailed.

### Ethics

Ethical approval was received from the "Comité d'Éthique de la Recherche Biomédicale" from the Ministry of Public Health in Madagascar (073-MSANP/CERBM) and the Institutional Review Board of Stony Brook University (CORIHS# 2017-4056-F). Written informed consent was obtained in the local Malagasy dialect from all participants.

## Results

### Participant characteristics

Fifteen participants were enrolled (see Table 1). The median age was 28 years (range: 19, 71 years) and seven (7/15) of the participants were male. Five (5/15) individuals were positive for active pulmonary tuberculosis (TB) at the time of data collection. All individuals were resident

**Table 1. Demographics, hourly PM$_{2.5}$ exposure levels, and hourly cough rates of participants.**

| ID | Sex | Age (years) | # people in household | Occupation | Smoking status | TB status | Follow-up (pro-rated hours) | Median (Q1, Q3) | Min, Max | Median (Q1, Q3) | Min, Max |
|---|---|---|---|---|---|---|---|---|---|---|---|
| | | | Demographic information | | | | | Hourly geometric mean PM$_{2.5}$ (µg/m³)[a] | | Hourly cough counts | |
| 1 | F | 19 | 7 | Farmer | Never | Neg. | 52 | 30.0 (6.4, 60.3) | 4.8, 388.8 | 18.5 (8, 30) | 0, 80 |
| 2 | F | 24 | 5 | Farmer | Never | Neg. | 16 | 8.8 (6.0, 16.0) | 1.0, 168.1 | 1.5 (0.5, 4) | 0, 38 |
| 3 | M | 57 | 8 | CHW | Never | Neg. | 20 | 7.7 (7.1, 32.7) | 0.7, 355.8 | 2 (1, 4) | 0, 21 |
| 4 | F | 23 | 3 | Farmer | Never | Neg. | 49 | 15.4 (12.0, 42.5) | 6.2, 179.7 | 10 (2, 24) | 0, 53 |
| 5 | F | 27 | 6 | Farmer | Never | Neg. | 48 | 23.1 (14.5, 48.4) | 10.5, 305.6 | 2.5 (0, 10) | 0, 29 |
| 6 | F | 29 | 14 | Farmer | Never | Pos. | 48 | 17.6 (10.3, 31.5) | 6.6, 123.1 | 3.5 (0.5, 7) | 0, 16 |
| 7 | M | 49 | 7 | Farmer | Never | Pos. | 54 | 22.6 (13.0, 39.2) | 2.3, 195.9 | 2 (0, 10) | 0, 45 |
| 8 | M | 71 | 11 | Farmer | Never | Pos. | 56 | 25.4 (17.4, 48.3) | 8.8, 156.1 | 3 (1, 14) | 0, 72 |
| 9 | F | 28 | 7 | Farmer | Never | Neg. | 49 | 10.8 (7.0, 25.7) | 3.8, 343.9 | 7 (2, 20) | 0, 94 |
| 10 | M | 25 | 7 | Farmer | Current smoker | Neg. | 49 | 14.1 (4.0,39.8) | 2.7, 613.3 | 5 (1, 13) | 0, 67 |
| 11 | F | 36 | 9 | Farmer | Never | Neg. | 48 | 39.8 (12.1, 76.9) | 4.1, 154.5 | 3 (0.5, 6.5) | 0, 26 |
| 12 | M | 28 | 10 | Farmer | Current smoker | Neg. | 45 | 12.5 (6.7, 39.0) | 4.3, 150.3 | 0 (0, 3) | 0, 8 |
| 13 | F | 48 | 8 | Farmer | Never | Neg. | 45 | 29.0 (11.1, 53.1) | 3.3, 232.8 | 7 (2, 19) | 0, 34 |
| 14 | M | 26 | 3 | Farmer | Smoked >5y ago | Pos. | 50 | 12.8 (10.1, 19.5) | 0.6, 214.0 | 1.5 (0, 8) | 0, 37 |
| 15 | M | 20 | 8 | Farmer | Smoked <5y ago | Pos. | 49 | 24.9 (11.5, 42.9) | 2.9, 190.5 | 0 (0, 6) | 0, 26 |
| Overall | M: 7/15 | Median: 28 | Median: 7 | Farmer: 14/15 | Never: 11/15 | Pos.: 5/15 | Total: 678 | 18.7 (10.3, 43.8) | 0.6, 613.3 | 4 (0, 11) | 0, 94 |
| | | Range: 19–71 | Range: 3–14 | | Current: 2/15 | | | | | | |
| | | | | | Smoked >5y: 2/15 | | | | | | |

F, female; M, male; CHW, community health worker; TB, tuberculosis; Neg., negative; Pos., positive; N, number of observations; SD, standard deviation; Q1, first quartile; Q3, third quartile.

[a] We took the geometric mean of the raw PM$_{2.5}$ measurements within each hour, and then reported the median of those hourly averages across the entire follow-up period.

in the Vatovavy region of Madagascar. All participants lived in traditional single or double room mud hut houses without ventilation systems. They all cooked indoors and relied primarily on charcoal or wood fire for cooking.

## PM$_{2.5}$ exposure and cough

A total of 678 person-hours of follow-up were collected from 15 participants. A total of 5,767 sounds were classified as cough, and 57 coughs were included as pro-rated coughs at the beginning and end of the recording session. The median coughs per hour was 4 coughs

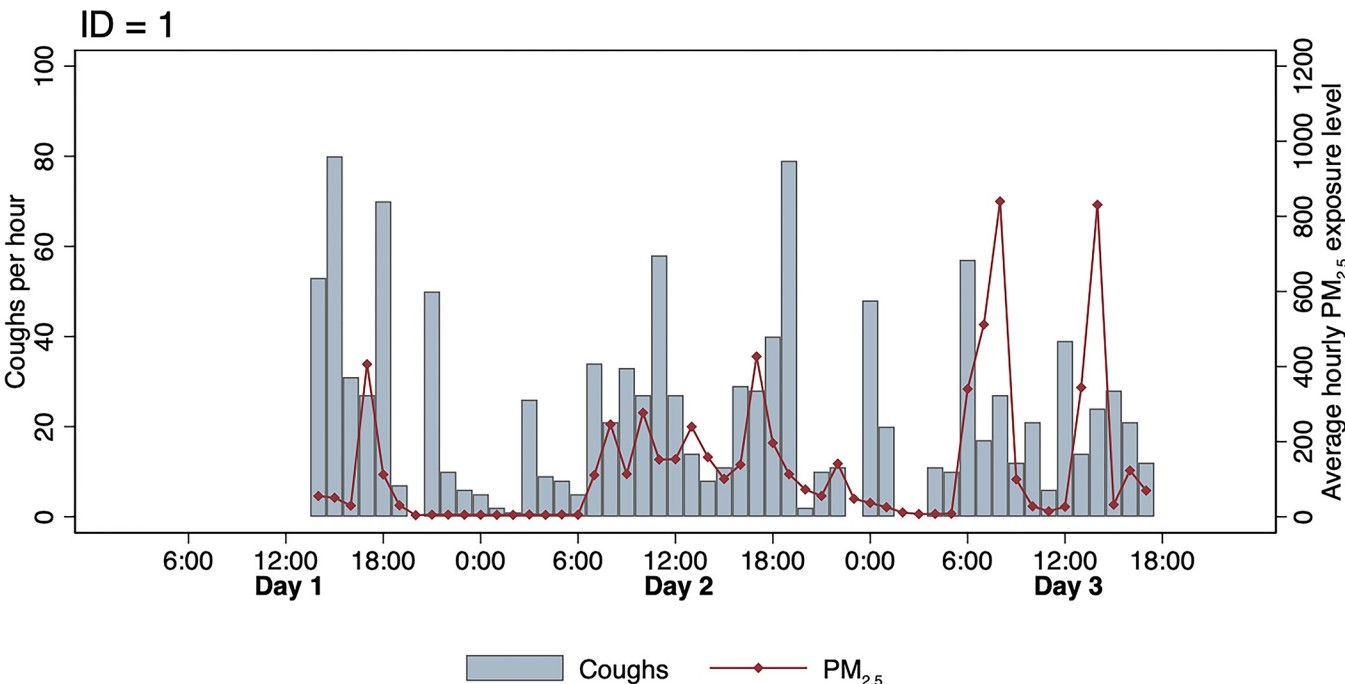

**Fig 1. Time series for participant ID = 1 showing the trends in number of coughs per hour and average hourly PM$_{2.5}$ exposure level.** Hourly PM2.5 concentrations and hourly cough counts appear to fluctuate throughout the day. Both measures tend to be highest during the daytime hours that coincide with cooking and mealtimes.

(interquartile range [IQR]: 0, 11). Hourly cough counts and personal exposure to PM$_{2.5}$ were overlaid for each participant (Figs 1 and S3).

### Association between hourly PM$_{2.5}$ concentrations and hourly cough counts

A 10-fold increase in hourly PM$_{2.5}$ concentrations was significantly associated with an increase in the population-average hourly cough rate with an incidence rate ratio (IRR) of 1.40 (95% confidence interval [CI]: 1.12, 1.74) at lag0 and an IRR of 1.21 (95% CI: 1.03, 1.46) at lag1 (Table 2, Models 1 and 2). This association was not significant when the lag between hourly PM$_{2.5}$ exposure and hourly cough count was increased to 2 hours (lag2) (IRR 0.93 [95% CI: 0.71, 1.23]) (Table 2, Model 3). When accounting for natural diurnal variations in cough frequency, the association between a 10-fold increase in PM$_{2.5}$ exposure and hourly cough count remained significant, though the effect appeared to be slightly attenuated with an IRR of 1.30 (95% CI: 1.04, 1.63) (Table 2, Model 4). This pattern of also occurred when incorporating hourly average temperatures into the model (Table 2, Model 5), resulting in an IRR of 1.38 (95% CI: 1.10, 1.72).

### Sensitivity analyses

Sensitivity analyses explored the effect of possible excess zeros resulting from participants potentially not carrying their phones to record cough sounds (S2 Fig). We found that the association between hourly PM$_{2.5}$ levels and cough frequency at lag0 remained unchanged. This observation was consistent when we excluded periods of 4h+, 6h+, and 8h+ of consecutive hours of zero cough recordings (Table 2, Models 6, 7, and 8). Finally, omitting raw PM$_{2.5}$ values measured above 15,000 μg/m$^3$ had no substantial effect on the results.

**Table 2. Negative binomial generalized estimated equation (GEE) on the association between hourly PM$_{2.5}$ exposure levels and coughs per hour.**

| Model | N | Hour lag | Variable | IRR (95% CI) | p-value |
|---|---|---|---|---|---|
| 1 | 678 | 0 | PM$_{2.5}$ (log$_{10}$) | 1.40 (1.12, 1.74) | 0.003 |
| 2 | 663 | 1 | PM$_{2.5}$ (log$_{10}$) | 1.21 (1.01, 1.44) | 0.034 |
| 3 | 648 | 2 | PM$_{2.5}$ (log$_{10}$) | 0.93 (0.71, 1.23) | 0.625 |
| 4 | 678 | 0 | PM$_{2.5}$ (log$_{10}$) | 1.30 (1.04, 1.63) | 0.019 |
| | | | *Day period interval* | | |
| | | | 0:00–5:59 | Ref. | Ref. |
| | | | 6:00–11:59 | 1.44 (1.04, 1.986) | 0.023 |
| | | | 12:00–17:59 | 1.25 (0.80, 1.95) | 0.323 |
| | | | 18:00–23:59 | 1.30 (0.95, 1.76) | 0.096 |
| 5 | 678 | 0 | PM$_{2.5}$ (log$_{10}$) | 1.38 (1.12, 1.70) | 0.003 |
| | | | Temperature (˚C) | 1.01 (0.98, 1.05) | 0.431 |
| 6 | 607 | 0 | PM$_{2.5}$ (log$_{10}$) | 1.38 (1.10, 1.72) | 0.005 |
| | | | *excl. 4h+ no coughs* | | |
| 7 | 633 | 0 | PM$_{2.5}$ (log$_{10}$) | 1.39 (1.12, 1.73) | 0.003 |
| | | | *excl. 6h+ no coughs* | | |
| 8 | 652 | 0 | PM$_{2.5}$ (log$_{10}$) | 1.37 (1.10, 1.70) | 0.005 |
| | | | *excl. 8h+ no coughs* | | |

N, number of hourly observations used in the analysis; IRR, incidence rate ratio; CI, confidence interval; excl., excluding.

Note: AR1 variance-covariance structure was used for all models.

## Stratified analyses

The association between hourly PM$_{2.5}$ levels and cough frequency at lag0 differed by smoking status and TB diagnosis (S1 Table). Past or current smokers coughed more frequently than individuals who never smoked, with an IRR of 2.21 (95% CI: 1.74, 2.80) compared to 1.22 (95% CI: 0.96, 1.55). Individuals with TB appeared to cough less frequently than individuals who were TB negative, with an IRR of 1.24 (95% CI: 0.95, 1.60) compared to 1.47 (95% CI: 1.11, 1.94).

## Discussion

Our investigation revealed high personal exposure to PM$_{2.5}$ among study participants, a finding that is congruent with prior literature documenting exposure to air pollution in Madagascar [18, 29]. Our analyses suggested an association between hourly cough counts and contemporaneous (same hour) PM$_{2.5}$ exposure levels. This association persists when factoring in a 1-hour lag between hourly PM$_{2.5}$ exposure and hourly cough counts; however, there is not clear trend when the lag is extended to 2 hours.

These findings suggest that individuals who are continuously exposed to elevated levels of air pollutants such as PM$_{2.5}$ could be at a greater risk of cough. This association could potentially confound the association between cough and other underlying causes, such as acute respiratory infections (e.g., TB) or chronic diseases (e.g., chronic obstructive pulmonary disease). The potential clinical implication is that individuals cannot easily recognize changes in cough frequency due to underlying medical conditions, delaying treatment and care. Further studies should investigate more in depth the impact of air pollution exposure on the reliability of cough-based clinical tools which are increasingly used to screen for or monitor respiratory disease.

Our study highlights the feasibility of collecting both personal exposure to PM$_{2.5}$ and objective cough counts in a remote setting such as rural Madagascar. The ResApp Cough Counting

Software application yielded comprehensive individual-level cough data despite being deployed in non-controlled acoustic environments and among participants with limited literacy and familiarity with the technology. The ECM sensor's ability to adjust to variable temperature and humidity in real time, as well as its ability to hold charge for several days, suggests that it is a reliable tool to measure $PM_{2.5}$ exposure within field conditions. Of note, changing filters in the field was occasionally challenging due to the lack of flat surfaces and optimal conditions with minimal air pollution exposure.

The small number of participants recruited in this study did not allow for multivariable analyses involving time invariant covariates, such as age (over a three-day period), sex, smoking status, and TB status. Stratified analyses were conducted according to levels of smoking status and TB diagnosis, two predictors known to influence cough frequency [14, 28]. However, these were exploratory and cannot be used to draw definite conclusions about how these predictors act as modifiers. The lower IRR among individuals with TB compared to those without TB may be attributed to the varying stages of anti-TB treatment among TB patients, which has been shown to reduce cough frequency within this population [30]. Data on individual patient treatment regimens was not available for this study. Thus, the analyses presented are intended to describe the association between personal exposure to $PM_{2.5}$ and cough counts and not establish causality. The one-to-three-day follow-up time may not be sufficient to meaningfully explore changes in cough frequency over time in relation to personal exposure to $PM_{2.5}$. Gabaldón-Figueira et al. found that individual cough patterns may be subject to stochastic variability within and between days [27]. Thus, to overcome the innate randomness in individual cough patterns, future studies should consider longer follow-up times. While the carrying case and instructions to keep the smartphone 25–50 cm away from their mouth was intended to minimize the recording of third-party cough sounds, it is possible that some of the cough sounds belong to someone other than the participant. This is a major limitation in the field of longitudinal cough monitoring.

In conclusion, our study underscores the relationship between $PM_{2.5}$ exposure and cough frequency, highlighting the potential implications for respiratory health and the need for more objective research on the acute impact of air pollution on respiratory symptoms and diseases. Further, this investigation demonstrates the feasibility of utilizing personal monitors, such as the ECM sensor and the smartphone-based cough monitor ResApp Cough Counting Software to effectively characterize exposure and cough counts in remote settings like rural Madagascar. Advancements in these monitoring technologies could aid in better understanding and addressing the effects of air pollution on respiratory health.

## Supporting information

**S1 Fig. Participant wearing the carrying pouch containing the iPhone 6 with the ResAppTM Health application and the MicroPEMTM monitoring devices.**
(DOCX)

**S2 Fig. Hourly cough recordings for each participant.**
(DOCX)

**S3 Fig. Individual cough and air pollution exposure monitoring for all 15 participants.**
(DOCX)

**S1 Table. Stratified analyses by TB status and smoking status for models with 0 hour lag between hourly PM2.5 exposure levels and coughs per hour.**
(DOCX)

## Acknowledgments

We thank all participants and their families for consenting to participate in an acoustic monitoring study design. We thank ResApp Health and RTI international for respectively providing the cough and air particle exposure monitoring devices.

## Author Contributions

**Conceptualization:** Lai Yu Tsang, Jesse McKinney, Simon Grandjean Lapierre.

**Data curation:** Alexandra J. Zimmer, Lai Yu Tsang, Gisèle Jolicoeur, Bouchra Tannir, Ryan Chartier.

**Formal analysis:** Alexandra J. Zimmer, Lai Yu Tsang, Alina Diachenko.

**Investigation:** Lai Yu Tsang, Christine Pando, Gouri Sadananda, Ideal Vincent Ambinintsoa, Roger Mario Rabetombosoa.

**Methodology:** Alexandra J. Zimmer, Lai Yu Tsang, Emmanuelle Batisse, Astrid M. Knoblauch, Ryan Chartier.

**Project administration:** Niaina Rakotosamimanana, Peter Small, Simon Grandjean Lapierre.

**Resources:** Niaina Rakotosamimanana, Peter Small, Simon Grandjean Lapierre.

**Software:** Gisèle Jolicoeur, Bouchra Tannir.

**Supervision:** Simon Grandjean Lapierre.

**Validation:** Alina Diachenko, Simon Grandjean Lapierre.

**Visualization:** Alexandra J. Zimmer, Lai Yu Tsang.

**Writing – original draft:** Alexandra J. Zimmer, Lai Yu Tsang.

**Writing – review & editing:** Simon Grandjean Lapierre.

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
