## [Decision Letter · Decision Letter 0]

17 May 2024

PGPH-D-24-00204

Incidence of cough from acute exposure to fine particulate matter (PM 2.5 ) in Madagascar: a pilot study

Dear Dr. Grandjean Lapierre,

Thank you for submitting your manuscript to PLOS Global Public Health. After careful consideration, we feel that it has merit but does not fully meet PLOS Global Public Health’s publication criteria as it currently stands. Therefore, we invite you to submit a revised version of the manuscript that addresses the points raised during the review process.

We look forward to receiving your revised manuscript.

Kind regards,

Naveen Puttaswamy, Ph.D

Academic Editor

Journal Requirements:

2. We noticed that you used "data not presented" in the manuscript. We do not allow these references, as the PLOS data access policy requires that all data be either published with the manuscript or made available in a publicly accessible database. Please amend the supplementary material to include the referenced data or remove the references.

Additional Editor Comments (if provided):

Reviewers' comments:

Reviewer's Responses to Questions

**Comments to the Author**

1. Does this manuscript meet PLOS Global Public Health’s publication criteria? Is the manuscript technically sound, and do the data support the conclusions? The manuscript must describe methodologically and ethically rigorous research with conclusions that are appropriately drawn based on the data presented.

Reviewer #1: Partly

Reviewer #2: Yes

2. Has the statistical analysis been performed appropriately and rigorously?

Reviewer #1: No

Reviewer #2: Yes

3. Have the authors made all data underlying the findings in their manuscript fully available (please refer to the Data Availability Statement at the start of the manuscript PDF file)?

Reviewer #1: Yes

Reviewer #2: No

4. Is the manuscript presented in an intelligible fashion and written in standard English?

Reviewer #1: Yes

Reviewer #2: Yes

5. Review Comments to the Author

Reviewer #1: The study is Novel Pilot study. This study can be described as a methods and feasibility study where the details of the process of measuring longitudinal simultaneous PM exposure and its acute effects (cough) are provided. Due to the small sample size, the strength of the association between PM and Cough is not scientifically valid. In addition, including two TB positive cases, smokers can be avoided as the sample size is very small. Hence can analyze the data by excluding the TB and smokers. If available more number of never smokers and non TB participants can be included.

Reviewer #2: The research described in the manuscript demonstrates methodological rigor by employing objective measures of PM2.5 exposure and cough frequency monitoring. The study design, including the use of univariable Generalized Estimating Equation (GEE) models, appears technically sound for investigating the association between PM2.5 exposure and respiratory symptoms. The conclusions drawn from the data presented are supported by the findings, which show a statistically significant association between PM2.5 exposure and cough frequency among study participants. Additionally, the manuscript acknowledges potential limitations, such as sample size constraints and confounding variables, and provides suggestions for future research to address these limitations. Overall, the manuscript meets the publication criteria of PLOS Global Public Health by presenting methodologically and ethically rigorous research with conclusions appropriately drawn based on the data presented.

Based on the provided author statement, the authors have made the code for cough screening publicly available on GitHub, which aligns with the requirement for data availability. However, the statement also mentions that the audio source file and PM 2.5 measurements datapoints are available upon request to the corresponding author, rather than being fully accessible without restriction. While this approach may be acceptable in some cases, it does not fully comply with the PLOS Data policy, which typically requires that all data underlying the findings described in the manuscript be made fully available without restriction, with rare exceptions. Therefore, the manuscript may not fully meet the data availability requirements of PLOS Global Public Health. To ensure compliance with the journal's publication criteria, the authors should consider providing a more comprehensive Data Availability Statement, specifying any restrictions on publicly sharing the data.

The manuscript is presented in an intelligible fashion and is written in standard English. The language used is clear and unambiguous, facilitating understanding of the research methods, results, and conclusions.

6. PLOS authors have the option to publish the peer review history of their article (what does this mean?). If published, this will include your full peer review and any attached files.

**Do you want your identity to be public for this peer review?** For information about this choice, including consent withdrawal, please see our Privacy Policy.

Reviewer #1: No

Reviewer #2: **Yes: **Rachit Sharma

---

## [Editor Report · Decision Letter 1]

5 Jul 2024

Incidence of cough from acute exposure to fine particulate matter (PM 2.5 ) in Madagascar: a pilot study

PGPH-D-24-00204R1

Dear Dr. Grandjean Lapierre,

We are pleased to inform you that your manuscript 'Incidence of cough from acute exposure to fine particulate matter (PM 2.5 ) in Madagascar: a pilot study' has been provisionally accepted for publication in PLOS Global Public Health.

Best regards,

Naveen Puttaswamy, Ph.D

Academic Editor